# SSL4EO-L:
# Datasets and Foundation Models for Landsat Imagery

**Adam J. Stewart[1], Nils Lehmann[2], Isaac A. Corley[3], Yi Wang[2, 4], Yi-Chia Chang[1],**
**Nassim Ait Ali Braham[2, 4], Shradha Sehgal[1], Caleb Robinson[5], Arindam Banerjee[1]**
[1]University of Illinois Urbana-Champaign,   [2]Technical University of Munich,
[3]University of Texas at San Antonio,   [4]German Aerospace Center,
[5]Microsoft AI for Good Research Lab

## Abstract

The Landsat program is the longest-running Earth observation program in history, with 50+ years of data acquisition by 8 satellites. The multispectral imagery captured by sensors onboard these satellites is critical for a wide range of scientific fields. Despite the increasing popularity of deep learning and remote sensing, the majority of researchers still use decision trees and random forests for Landsat image analysis due to the prevalence of small labeled datasets and lack of foundation models. In this paper, we introduce SSL4EO-L, the first ever dataset designed for **S**elf-**S**upervised **L**earning for **E**arth **O**bservation for the **L**andsat family of satellites (including 3 sensors and 2 product levels) and the largest Landsat dataset in history (5M image patches). Additionally, we modernize and re-release the L7 Irish and L8 Biome cloud detection datasets, and introduce the first ML benchmark datasets for Landsats 4–5 TM and Landsat 7 ETM+ SR. Finally, we pre-train the first foundation models for Landsat imagery using SSL4EO-L and evaluate their performance on multiple semantic segmentation tasks. All datasets and model weights are available via the TorchGeo[1] library, making reproducibility and experimentation easy, and enabling scientific advancements in the burgeoning field of remote sensing for a multitude of downstream applications.

## 1   Introduction

On July 23$^{rd}$, 1972, the National Aeronautics and Space Administration (NASA) launched Landsat 1. Designed by Virginia T. Norwood, the Multispectral Scanner (MSS) onboard Landsats 1–5 provided invaluable measurements of the Earth's surface in both the visible and infrared spectra. Although she passed away earlier this year, her legacy as "The Mother of Landsat" lives on, as the Landsat program has become the longest-running Earth observation program in history [1].

The Landsat satellite program stretches over 50 years and includes 9 generations of satellites, each with its own set of *sensors*. Landsats 1–3 carried the Return Beam Vidicon (RBV), an RGB analog camera [2]. However, its lower number of spectral bands and electrical issues meant it was rarely used for research purposes. Instead, the Multispectral Scanner (MSS) onboard Landsats 1–5 was the primary scientific instrument, with a line-scanning and rotating mirror-based camera [3]. Landsats 4–5 also included the Thematic Mapper (TM), with a greater number of spectral bands (from 4 to 7) and finer spatial resolution (from 80 m to 30 m) [4]. Although it failed to reach orbit and eventually crashed down to Earth, Landsat 6 would have carried the Enhanced Thematic Mapper (ETM), which added a 15 m resolution panchromatic band. Landsat 7 carried the Enhanced Thematic Mapper Plus

---

[1]https://github.com/microsoft/torchgeo

37th Conference on Neural Information Processing Systems (NeurIPS 2023) Track on Datasets and Benchmarks.

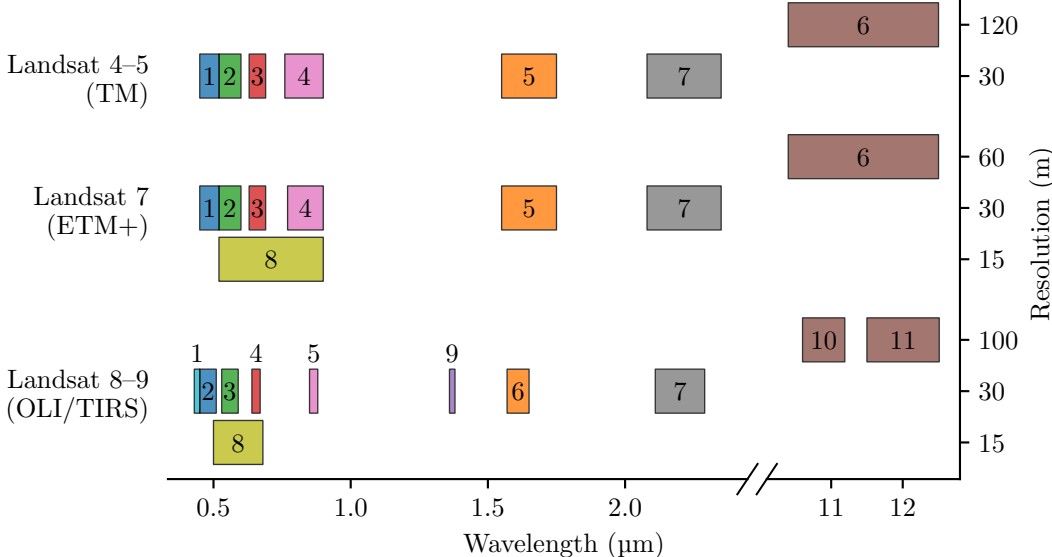

Figure 1: Spectral wavelengths and spatial resolutions of each band captured by the Landsat sensors used in our study. Numbers are band indices and colors are for visualization purposes only. As each sensor has a different number of spectral bands, spatial resolutions, and wavelengths, it is not possible to train a single one-size-fits-all model.

(ETM+), which upgraded the thermal band from 120 m to 60 m resolution [5]. Onboard Landsats 8–9, the Operational Land Imager (OLI) adds new coastal aerosol and cirrus bands for improved cloud masking, while the Thermal Infrared Sensor (TIRS) adds an additional thermal band [6]. See Figure 1 for a rundown of the spectral wavelengths and spatial resolutions of the primary sensors of interest in our study and Figure 2 for a timeline of when these satellites were active.

In addition to the differences between sensors onboard each satellite, the United States Geological Survey (USGS) distributes several different Landsat *products* with varying processing levels. Level-1 data, also known as Top of Atmosphere (TOA), are images that have undergone registration against ground control points (GCPs) and orthorectification against digital elevation models (DEMs). These products are particularly useful for cloud masking and other atmospheric applications. Level-2 data includes Surface Reflectance (SR) and other products that have undergone atmospheric correction. These products are useful for a wide range of land surface applications [7].

In recent years, there has been significant activity at the intersection of self-supervised learning (SSL) and remote sensing (RS) due to the wide availability of petabytes of free, unlabeled satellite imagery. An early example of this is Tile2Vec [8], which uses geographic distance between sampled patches and a triplet margin loss for contrastive learning. Geography-Aware SSL [9] instead uses multiple images occurring at the *same* geospatial location at *different* points in time to form positive pairs, in combination with an additional subnetwork that tries to guess the latitude/longitude from the learned representation. More recently, masked autoencoders have seen a surge in popularity, including SatMAE [10] and Scale-MAE [11]. Other papers instead focus on dataset curation, allowing generic SSL techniques from computer vision to be applied. Seasonal Contrast (SeCo) [12] creates a new dataset using random Gaussian sampling around cities to diversify the pre-training dataset. SSL4EO-S12 [13] further extends this idea by avoiding overlap between image samples.

Recent review papers [14, 15, 16] targeting the intersection between SSL and RS detail prior work on this topic and offer benchmark results on several models and SSL techniques popular in computer vision, including MoCo [17], SwAV [18], SimSiam [19], Barlow Twins [20], SimCLR [21], and BYOL [22]. Among these, the authors find that although SimCLR and SwAV work well on the ImageNet [23] dataset, MoCo and BYOL tend to learn better representations on RS imagery [15, 16].

Due to their higher spatial resolution and faster repeat period, a lot of recent work, especially in the SSL space, focuses on Sentinel-2 [24, 25, 13], Maxar [9, 26, 27], and Planet [28, 29] satellites. However, many applications are not suited to these satellites. In particular, applications involving

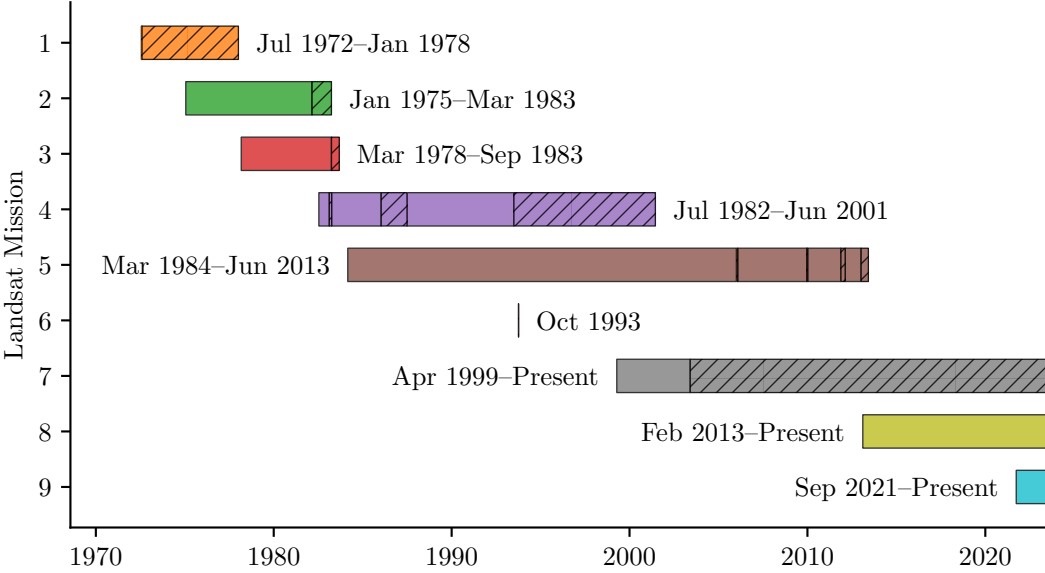

Figure 2: Timeline of all Landsat missions. Crosshatched regions represent partial or complete sensor failure, including electrical issues with RBV on Landsat 1 and SLC-off on Landsat 7. Each bar ranges from launch date to decommissioning date. Note that satellites may be placed in standby mode before the decommissioning date, as is the case with Landsat 4.

long-term trends—including agriculture [30, 31, 32, 33, 34], climate change [35, 36, 37, 38, 39], deforestation [40, 41, 42, 43, 44], and ecology [45, 46, 47, 48, 49]—require a much longer temporal history. While Sentinel-2 has an 8 year history, Landsat's 50+ year history makes it essential for monitoring long-term land surface changes. There are an order of magnitude more papers that use Landsat as compared to Sentinel, and Landsat continues to dominate the scientific literature even after the launch of Sentinel-2 and MODIS [50]. The United States Geological Survey (USGS) estimates that Landsat imagery provides users with an annual benefit of $4.2 billion [51].

In this work, we further extend the ideas proposed in SeCo [12] and SSL4EO-S12 [13] to the Landsat imagery domain. Specifically, we improve on the sampling method of the two previously mentioned papers and sample Landsat imagery across the world and across three different *sensors* and two *product* levels. We pre-train ResNet [52] and ViT [53] models using SimCLR v1 and MoCo v2 on each combination of sensor and product to produce a suite of pre-trained Landsat foundation models that can be used in downstream tasks. To test these models, we modernize two older datasets based on Landsat 7 and 8 imagery, and create several additional crop classification and land cover mapping tasks. In summary, the contributions of this paper include:

- the first ever SSL dataset for the Landsat family of satellites,
- the largest Landsat dataset in history (1M images per sensor/product, 5M in total),
- modernized and re-released versions of the L7 Irish and L8 Biome cloud detection datasets,
- two new benchmark datasets that can be used across all Landsat sensors and product levels,
- the first ever benchmark datasets for TM and ETM+ SR imagery, and
- the first ever foundation models pre-trained on Landsat imagery.

Importantly, all of these SSL techniques, datasets, and pre-trained models are distributed via the TorchGeo library [54], allowing for ease of use, experimentation, and reproducibility.

## 2 Datasets

In this section, we detail our methodology behind the collection of the SSL dataset we create, including differences from prior work. We also introduce the existing and newly created benchmark datasets we use to evaluate the representations learned by our pre-trained models.

## 2.1 SSL4EO-L pre-training dataset

For our SSL pre-training dataset, we extend the methodology introduced by Manas et al. [12] and refined by Wang et al. [13]. Specifically, we iterate over the following steps:

1) sample one of the 10K most populous cities in the world [55] uniformly at random;

2) sample a $264 \times 264$ px ($7.92 \times 7.92$ km) patch from a Gaussian distribution with a 50 km standard deviation centered around the centroid of the sampled city;

3) ensure the patch does not overlap with any existing sampled patches;

4) ensure that there exist 4 patches of imagery from 4 different seasons—each selected from a 60-day window centered about the vernal and autumnal equinoxes and the summer and winter solstices (within a 2-year window)—with less than 20% cloud coverage;

5) ensure that none of these patches contain nodata pixels;

6) if the previous three criteria are met, download the imagery corresponding to the patch.

If any step in this algorithm fails (there is overlap, or a location does not have a set of 4 cloud-free, nodata-free images), the sample is skipped and we start over at step 1. This algorithm is designed to maximize the diversity of images in the dataset, relying on the assumption that most of the diversity in land cover is centered around large cities, with a gradual transition between urban, suburban, farmland, and forest. Uniform sampling would instead result in images that are 70% ocean, 10% desert, and 9% forest, resulting in very little dataset diversity [12]. Note that this sampling strategy does result in decreased sampling from regions with persistent cloud cover (tropical rainforests) or lower populations (desert, taiga, tundra, and polar biomes). By sampling different points in time, we allow seasonal differences to act as natural forms of data augmentation during contrastive learning.

Differences between our sampling strategy and the one used by SSL4EO-S12 are as follows. SSL4EO-S12 used Euclidean distance between patch centroids and a grid heuristic to detect overlap between patches. This method has an $\mathcal{O}\left(N^2/M\right)$ average run-time complexity, where $N$ is the total number of samples and $M$ is the number of grid cells. We replace this with an $\mathcal{O}\left(N \log N\right)$ R-tree [56], removing the 1–3% overlap reported by Wang et al. [14] due to use of this grid heuristic. Among the cloud-free images in the aforementioned time windows, we sort by cloud cover instead of date to provide the best possible image patches. We also skip patches containing nodata pixels due to sampling near the border of a scene, which we found to be prevalent (on the order of 25%) in prior datasets. We found it necessary to increase the cloud coverage threshold from 10% to 20% due to the larger patch size (Sentinel-2 has a 10 m resolution, but Landsat has a 30 m resolution, resulting in patches that cover 9× the area) and avoidance of nodata pixels. Finally, since the resolution of most bands are the same, we resample all thermal and panchromatic bands to a 30 m resolution, allowing all bands to be concatenated into a single file.

We download all data from Google Earth Engine (GEE) [57], with a total of 250K locations, each sampled at 4 different seasons, for a total of 1M unlabeled image patches per sensor/product and 5M in total. Each image is $264 \times 264$ px, corresponding to $7.92 \times 7.92$ km at 30 m/px resolution. There are separate datasets for TM TOA, ETM+ TOA, ETM+ SR, OLI/TIRS TOA, and OLI SR. We decided not to include RBV and MSS sensors due to the limited data availability on GEE and the fact that it is not possible to create a benchmark dataset for these sensors due to their age. Since TM and ETM+ use the same sensor for SR bands, we did not create a separate dataset for TM SR. For similar reasons, there is a single dataset for OLI/TIRS and OLI-2/TIRS-2. TM data is collected from 4 different seasons in 2009–2010, as the TM sensor failed in November 2011. ETM+ data is collected from 2001–2002, as the scan line corrector (SLC) failed in May, 2003, resulting in images with significant nodata pixels. OLI/TIRS data is collected from 2021–2022. See Figure 3 for a map of the geographical distribution for each sensor. Note that it is not possible to sample high latitudes due to lack of winter imagery.

All TOA and SR datasets represent a parallel corpus (the TOA and SR images are taken at the same locations and dates). Due to differences in collection years and cloud coverage/nodata pixels, it was not possible to create a parallel corpus between sensors. However, approximately 50% of TM and ETM+, 40% of TM and OLI/TIRS, and 40% of ETM+ and OLI/TIRS images are sampled from the same location, allowing for multimodal data fusion studies. The official scale factors suggested

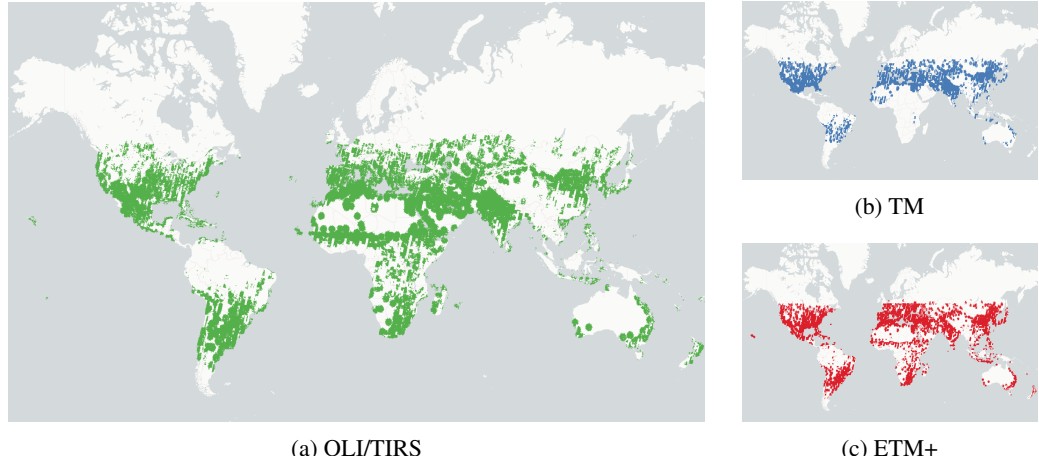

(a) OLI/TIRS

(b) TM

(c) ETM+

Figure 3: Geographical distribution of the SSL4EO-L dataset, including the (a) Landsat 8–9 OLI/TIRS, (b) Landsat 4–5 TM, and (c) Landsat 7 ETM+ splits. Surface reflectance (SR) and top of atmosphere (TOA) products are sampled from the same locations per sensor.

by the USGS to map between Level-1 and Level-2 Landsat imagery[2] and the visualization range recommended by GEE for each sensor are used to map from float32 to uint8. The resulting datasets are 274–385 GB when compressed and can be downloaded from Hugging Face[3] using TorchGeo.

## 2.2 Dataset archaeology

In order to benchmark the ability of our learned representations to transfer to downstream applications, we require curated benchmark datasets for evaluation. Although there exist ∼ 10 semantic segmentation datasets for OLI/TIRS TOA, an extensive literature review found almost no benchmark datasets for other sensors, products, or tasks. This is due to both their age (deep learning was not commonplace in the field of remote sensing until recently) and the fact that semantic segmentation is the primary task for which lower resolution satellite imagery is used.

A single classification dataset, Statlog [58], was found for the MSS sensor. However, this dataset is composed of $3 \times 3$ px images, making it unsuitable for evaluation of CNN and ViT backbones. For the task of semantic segmentation for cloud cover, three ETM+ TOA datasets were found: L7 SPARCS [59], L7 Irish [60, 61], and L7 Scaramuzza [62]. Each of these datasets also has a corresponding dataset for OLI/TIRS TOA (L8 SPARCS [63, 64], L8 Biome [65, 66], and L8 Scaramuzza [67]), making it possible to compare learned representations across sensors. No benchmark datasets for TM or ETM+ SR were ever found. The L7 SPARCS dataset, while thought to be lost to time, was eventually recovered from a hard drive found in the closet of one of the dataset's authors. The majority of the aforementioned cloud segmentation datasets are official datasets used by the USGS to validate their cloud detection algorithms. Among these datasets, we chose to use L7 Irish and L8 Biome due to their larger size and greater number of citations.

### 2.2.1 L7 Irish dataset

The L7 Irish dataset, originally selected by Irish et al. [68] and later digitized by Scaramuzza et al. [61], is a validation dataset for cloud cover assessment algorithms composed of 206 Landsat 7 ETM+ Level-1G scenes and manually generated cloud masks divided between 9 unique biomes. Each scene is a 9-band, roughly $8000 \times 8000$ px multispectral image with 30 m/px resolution. Cloud masks consist of 5 classes: 1) fill, 2) cloud shadow, 3) clear, 4) thin cloud, and 5) cloud.

There are 2015 [69] and 2019 [60] versions of this dataset available for download. Unfortunately, both versions have numerous issues that make them difficult to use for evaluation. The 2015 version contains 1 scene with a corrupted thermal band file, 2 scenes that are missing masks, 1 scene with an

[2]https://www.usgs.gov/faqs/how-do-i-use-scale-factor-landsat-level-2-science-products
[3]https://huggingface.co/torchgeo

inconsistent filename format, and the documented class labels do not match the actual class labels used. Additionally, there is no way to programmatically download the entire dataset. All 206 files must be manually downloaded, one at a time, with a limit of 6 parallel downloads, requiring 3–4 hrs of constant supervision and clicking each link every 5 min. The 2019 version has even more issues, including 5 scenes with corrupted thermal band files, 1 scene missing geolocation, 6 scenes with inconsistent filename formats, and inconsistent thermal band resolutions. Although 17% of masks matched the documented labels, the other 83% of masks use a completely different mapping, with both clear and fill classes mapped to the same value.

In order to use this dataset for evaluation, we start with the 2015 version and use scenes from the 2019 version to replace corrupted images and missing masks. We correct the class mapping of copied masks and copy the fill pixels from the images to the masks. We convert all images to Cloud Optimized GeoTIFFs (COGs), resample to 30 m resolution, and stack them into single multi-band files with consistent filenames. The compression algorithm used by COGs resulted in a dataset that is 33% of the original size and therefore faster to download and load from disk. The final ML-ready dataset is available on Hugging Face and can be automatically downloaded using TorchGeo.

### 2.2.2 L8 Biome dataset

The L8 Biome dataset, created by Foga et al. [65], is a validation dataset for cloud cover assessment algorithms consisting of 96 Landsat 8 OLI/TIRS Level-1T scenes and manually generated cloud masks evenly divided between 8 unique biomes. Each scene is an 11-band, roughly $9000 \times 9000$ px multispectral image with 30 m/px resolution. Cloud masks consist of the same 5 classes as L7 Irish.

Comparatively, L8 Biome has fewer issues than L7 Irish. The masks lack geolocation, but we can copy this from the image files. While the dataset can be programmatically downloaded, it requires scraping a webpage for 96 different URLs for each scene. We convert the raw uint16 images to uint8 to match L7 Irish, and create compressed COGs of all files, resulting in a dataset 9% of the original size. We resample all images to 30 m/px resolution and stack them in single multi-band files. The dataset is available on Hugging Face and can be automatically downloaded using TorchGeo.

### 2.3 SSL4EO-L benchmark dataset

As there are no existing benchmark datasets for TM or ETM+ SR, we need to design our own. Crucially, we want a single benchmark dataset that can be used for a consistent comparison across all 5 sensors/products for which we are pre-training models. We create our own land cover classification datasets based on NLCD [70] and CDL [71] masks, described in more detail below. They are the only large, Landsat-based semantic segmentation masks with a long enough history to benchmark foundation models for historical satellites.

Our sampling strategy is similar to the one used for our pre-training dataset, with a few differences. As CDL only exists for the continental U.S. (CONUS), we restrict our sampling strategy to CONUS. To achieve maximum coverage, especially in lower population regions where agriculture is most prevalent, we replace the city-centered Gaussian distribution with a uniform sampling distribution. We choose a single 60-day window centered around August $1^{st}$ when crop types are easiest to distinguish. As CDL data is not available before the ETM+ SLC failure, we do not exclude no-data pixels for this sensor. Additionally, nodata masks are copied from SLC-off imagery to masks so as to avoid penalizing models for making incorrect predictions where there is no data. The 2019 NLCD and CDL datasets are used for ETM+ and OLI/TIRS evaluation since 2019 is the most recent year for which both datasets exist. The 2011 datasets are used for TM since 2011 is the most recent year for which both Landsat 5 and NLCD/CDL overlap. These years are different than the years collected for our pre-training dataset, allowing us to accurately measure performance on images that the pre-trained model has never seen before.

The resulting dataset consists of 25K Landsat, NLCD, and CDL triplets, converted from float32 to uint8 using the same scaling as above. All images have the same resolution and dimensions as the pre-training dataset. The datasets form a parallel corpus between TOA and SR products, and have approximately 85% spatial overlap across sensors, although not necessarily during the same year, allowing for multimodal data fusion studies. All datasets are available for download from Hugging Face using the TorchGeo library, making it easy for other researchers to compare against our preliminary benchmark results.

**NLCD**    The National Land Cover Database (NLCD) [70] is a land cover product produced every 2–3 years by the USGS, in collaboration with the Multi-Resolution Land Characteristics (MRLC) consortium. The dataset spans the entire U.S. from 2001–2019. The final products are generated at a 30 m resolution by random forest models trained on spectral, spatial, temporal, and ancillary data [72, 73, 74]. We use the 21 class version, with an estimated overall accuracy of 77.5±1.0% [75].

**CDL**    The Cropland Data Layer (CDL) [32] is an annual land cover product produced by the U.S. Department of Agriculture (USDA) National Agricultural Statistics Service (NASS) focusing on crop classification. Although the dataset is available starting in 1997, full CONUS coverage is not available until 2008. The dataset consists of 134 classes, primarily for agricultural crops grown in the U.S. Labels are generated at a 30 m resolution using a decision tree classifier. The most common crop classes are estimated to have an accuracy of 85–95% [32]. All non-agricultural classes are taken from NLCD, and should be considered to have a similar accuracy.

## 3    Experimental setup

For pre-training we conduct experiments similar to those performed in SSL4EO-S12 [13] for each sensor/product in the dataset described in Section 2.1. We pre-train various ResNet [52] and ViT [53] backbones initialized with ImageNet weights using the SimCLR v1 [21] and MoCo v2 [17] SSL methods. RGB ImageNet weights are repeated (RGBRGB...) and scaled ($3/C$ for $C$ channels) in the first convolutional layer in order to handle multispectral images. During pre-training we use the same default augmentations and hyperparameters as SimCLR and MoCo with a couple of exceptions. As saturation and hue are undefined for multispectral imagery, we skip these parts of color jitter. Instead, we use the random season contrast technique proposed by Manas et al. [12] by utilizing 2 randomly sampled multitemporal images from the same location as the augmented views. Additionally, although grayscale is undefined for multispectral imagery, we take the average of all bands to compute random grayscale images. We pre-train each model for 200 epochs using a batch size of 1024. All pre-training experiments are performed on a GPU cluster, with 80 GB of memory per GPU. Each experiment takes anywhere from 15–40 hrs depending on the number of spectral bands and model size, each trained in parallel on 4× GPUs, for a total of ∼4K GPU hours including hyperparameter tuning.

For benchmarking, we freeze the encoder and fine-tune a U-Net [76] decoder for all cloud detection and land cover classification datasets mentioned above. For the L7 Irish and L8 Biome datasets, we use a random 60-20-20 train-val-test split. For the NLCD and CDL datasets, we use a random 70-15-15 train-val-test split. NLCD and CDL classes are limited to those with >1% area, with remaining classes mapped to the background class. Splits are defined using a fixed random seed for reproducibility. Random horizontal and vertical flip and random resized crop data augmentations are used during training. Models are trained for a minimum of 20 epochs and a maximum of 100 epochs using early stopping and a learning rate schedule patience of 6 epochs. Only learning rate undergoes hyperparameter tuning, with the most common optimal learning rate being 3e-3. All benchmarking experiments are conducted on NVIDIA RTX A6000 (2.5 hr/experiment) and A100 (1 hr/experiment) GPUs for a total of ∼200 GPU hours. Configuration files and training scripts for reproducing all experiments are made available in the TorchGeo library [54].

## 4    Benchmark results

In order to evaluate the effectiveness of our pre-trained models, we report overall accuracy and mean intersection over union (mIoU) on four semantic segmentation datasets. Table 1 demonstrates substantial gains over ImageNet, with up to an 18.43% accuracy and 24.25 mIoU improvement for MoCo and up to a 14.43% accuracy and 18.69 mIoU improvement for SimCLR. Although MoCo outperforms ImageNet in 5 out of 6 experiments, SimCLR shows mixed results, outperforming ImageNet in only 2 out of 6 experiments. Our SimCLR models suffered from convergence issues with the smaller batch size we used, and may improve with better hyperparameter tuning.

Note that both our sampling method and pretext task are explicitly designed to ignore clouds. During sampling, we only select patches from scenes with <20% cloud cover, decreasing the frequency of clouds in our pre-training dataset. Our pretext task involves mapping patches taken from 2 different seasons to the same representation. If one patch contains partial cloud cover, the model must learn to

Table 1: Cloud detection benchmark results. Overall accuracy and mean intersection over union (mIoU) are reported for the test splits of the L7 Irish (Landsat 7 ETM+ TOA) and L8 Biome (Landsat 8 OLI/TIRS TOA) datasets for a range of backbones and pre-training techniques. All predictions are made by U-Nets with frozen backbones. Three random seeds are used to compute mean ± standard deviation of the performance.

| | | L7 Irish | | L8 Biome | |
|---|---|---|---|---|---|
| Backbone | Pre-training | Accuracy | mIoU | Accuracy | mIoU |
| ResNet-18 | ImageNet | 64.08 ± 3.40 | 47.21 ± 3.71 | 41.86 ± 0.46 | 24.67 ± 0.37 |
| | MoCo | **74.79 ± 2.20** | **59.77 ± 2.79** | **42.70 ± 5.02** | **27.33 ± 4.14** |
| | SimCLR | 34.80 ± 11.36 | 21.46 ± 8.53 | 39.17 ± 5.12 | 24.44 ± 3.89 |
| ResNet-50 | ImageNet | 61.77 ± 3.27 | 44.75 ± 3.41 | 45.73 ± 6.08 | 29.78 ± 5.23 |
| | MoCo | **69.62 ± 1.94** | **53.42 ± 2.29** | 45.95 ± 5.17 | 29.23 ± 4.44 |
| | SimCLR | 49.37 ± 12.86 | 33.41 ± 11.20 | **48.77 ± 6.42** | **32.41 ± 5.56** |
| ViT-S16 | ImageNet | 68.22 ± 1.39 | 51.78 ± 1.59 | **47.29 ± 1.88** | **30.98 ± 1.61** |
| | MoCo | **86.65 ± 0.43** | **76.03 ± 0.67** | 46.66 ± 3.59 | 30.33 ± 3.14 |
| | SimCLR | 82.65 ± 0.27 | 70.47 ± 0.35 | 42.33 ± 1.80 | 26.99 ± 1.67 |

ignore it. The fact that our SSL techniques work at all, let alone outperform ImageNet, demonstrates the generalizability of our pre-trained model weights to different downstream applications.

Performance metrics for our land cover/land use tasks are reported in Table 2. Again, MoCo consistently outperforms ImageNet in 25 out of 30 experiments across all sensors and product levels, while SimCLR is unable to beat ImageNet in 24 out of 30 experiments. Performance gains by MoCo are more modest in this task with a larger number of classes, but still reach as high as 6.60% overall accuracy and 7.13 mIoU. There are exceptions to this, particularly for ETM+ TOA, but with additional hyperparameter tuning of the pre-trained model it may be possible to exceed performance of ImageNet. We attempted to use weights based on class frequency in our cross-entropy loss, but these resulted in reduced accuracy and mIoU.

Figure 4 shows an example prediction made by a ResNet-18 backbone pre-trained on SSL4EO-L using MoCo and a U-Net fine-tuned on CDL. Although the model is unable to predict detailed features like roads and field corners, it removes much of the noise introduced by the pixel-wise decision tree classifier used to produce CDL. Black pixels in the mask represent uncommon crop types that are mapped to the background class. The model tends to pick the most common agricultural classes like corn and soybean given no examples of these crop types in the training dataset. Although winter wheat and fallow (idle farmland) are sometimes misclassified by the model, this is not unexpected.

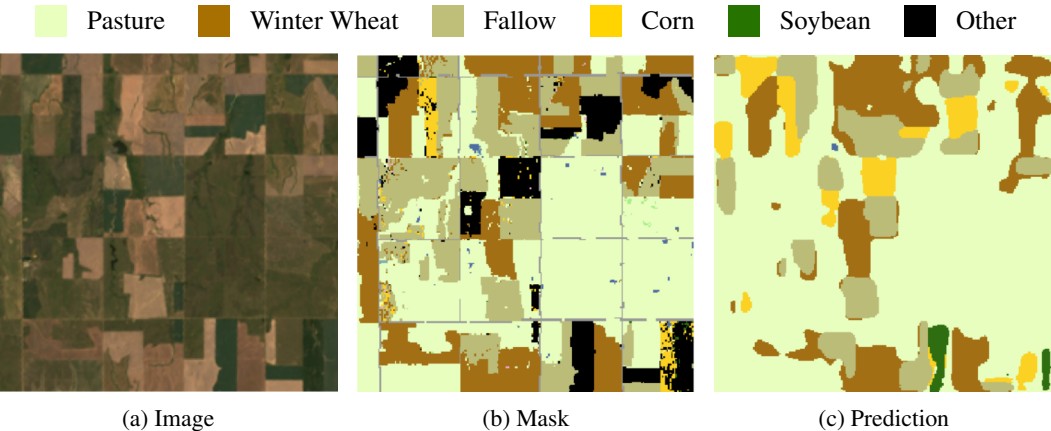

(a) Image          (b) Mask          (c) Prediction

Figure 4: Landsat 8 OLI SR image, ground truth mask, and prediction made by a U-Net with a ResNet-18 backbone pre-trained using MoCo and SSL4EO-L and fine-tuned on CDL 2019.

Table 2: SSL4EO-L benchmark results. Overall accuracy and mean intersection over union (mIoU) are reported for the test splits of the NLCD and CDL datasets for a range of sensors, product levels, backbones, and pre-training techniques. All predictions are made by U-Nets with frozen backbones.

| Satellite (Sensor) | Level (Product) | Backbone | Pre-training | NLCD Accuracy | NLCD mIoU | CDL Accuracy | CDL mIoU |
|---|---|---|---|---|---|---|---|
| Landsats 4–5 (TM) | Level-1 (TOA) | ResNet-18 | ImageNet | 65.63 | 48.84 | 66.11 | 49.38 |
| | | | MoCo | **67.65** | **51.11** | **68.70** | **52.32** |
| | | | SimCLR | 60.86 | 43.74 | 61.94 | 44.86 |
| | | ResNet-50 | ImageNet | 66.63 | 49.96 | 67.42 | 50.85 |
| | | | MoCo | **68.75** | **53.28** | **69.45** | **53.20** |
| | | | SimCLR | 62.05 | 44.98 | 62.80 | 45.77 |
| | | ViT-S16 | ImageNet | **68.93** | **52.59** | **68.27** | **51.83** |
| | | | MoCo | 67.17 | 50.57 | 67.60 | 51.07 |
| | | | SimCLR | 66.82 | 50.17 | 66.92 | 50.28 |
| Landsat 7 (ETM+) | Level-1 (TOA) | ResNet-18 | ImageNet | **66.11** | **49.38** | **65.84** | **49.08** |
| | | | MoCo | 65.22 | 48.39 | 62.84 | 45.81 |
| | | | SimCLR | 58.76 | 41.60 | 56.47 | 39.34 |
| | | ResNet-50 | ImageNet | 64.01 | 47.06 | **66.23** | **49.51** |
| | | | MoCo | **66.60** | **49.92** | 64.12 | 47.19 |
| | | | SimCLR | 57.17 | 40.02 | 54.95 | 37.88 |
| | | ViT-S16 | ImageNet | 62.06 | 45.01 | 57.67 | 40.52 |
| | | | MoCo | **63.75** | **46.79** | **60.88** | **43.70** |
| | | | SimCLR | 63.33 | 46.34 | 59.06 | 41.91 |
| | Level-2 (SR) | ResNet-18 | ImageNet | 63.34 | 46.34 | 60.70 | 43.58 |
| | | | MoCo | **64.18** | **47.25** | **67.30** | **50.71** |
| | | | SimCLR | 57.26 | 40.11 | 54.42 | 37.48 |
| | | ResNet-50 | ImageNet | 64.29 | 47.38 | 61.66 | 44.57 |
| | | | MoCo | **64.37** | **47.46** | **62.35** | **45.30** |
| | | | SimCLR | 57.79 | 40.64 | 55.69 | 38.59 |
| | | ViT-S16 | ImageNet | 63.54 | 46.56 | 51.38 | 34.57 |
| | | | MoCo | **64.09** | **47.21** | 52.37 | 35.48 |
| | | | SimCLR | 63.99 | 47.05 | **53.17** | **36.21** |
| Landsats 8–9 (OLI/TIRS) | Level-1 (TOA) | ResNet-18 | ImageNet | 66.40 | 49.70 | 65.21 | 48.38 |
| | | | MoCo | **67.82** | **51.30** | **65.74** | **48.96** |
| | | | SimCLR | 62.14 | 45.08 | 60.01 | 42.86 |
| | | ResNet-50 | ImageNet | 67.73 | 51.20 | 66.45 | 49.76 |
| | | | MoCo | **69.17** | **52.87** | **67.29** | **50.70** |
| | | | SimCLR | 64.66 | 47.78 | 62.08 | 45.01 |
| | | ViT-S16 | ImageNet | 65.52 | 48.72 | 62.38 | 45.33 |
| | | | MoCo | **67.11** | **50.49** | **64.62** | **47.73** |
| | | | SimCLR | 66.12 | 49.39 | 63.88 | 46.94 |
| | Level-2 (SR) | ResNet-18 | ImageNet | 65.46 | 48.65 | 62.88 | 45.85 |
| | | | MoCo | **67.01** | **50.39** | **68.05** | **51.57** |
| | | | SimCLR | 59.93 | 42.79 | 57.44 | 40.30 |
| | | ResNet-50 | ImageNet | 66.29 | 49.58 | 64.17 | 47.24 |
| | | | MoCo | **67.44** | **50.88** | **65.96** | **49.21** |
| | | | SimCLR | 63.65 | 46.68 | 60.01 | 43.17 |
| | | ViT-S16 | ImageNet | 65.71 | 48.93 | 62.78 | 45.75 |
| | | | MoCo | **66.81** | **50.16** | **64.17** | **47.24** |
| | | | SimCLR | 65.04 | 48.20 | 62.61 | 45.46 |

Winter wheat is planted in the fall and may be harvested before our summer imagery is taken. Similarly, fallow can look much like pasture as weeds begin to grow in empty fields.

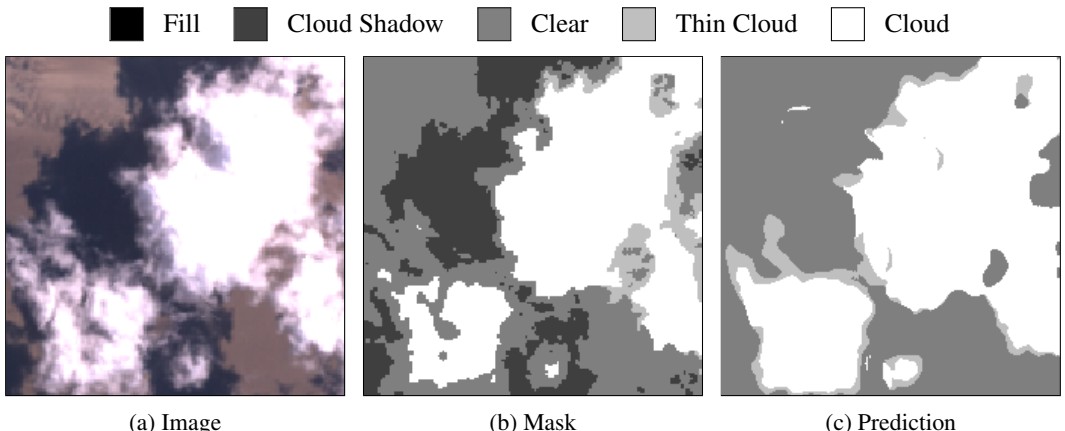

(a) Image        (b) Mask        (c) Prediction

Figure 5: Landsat 7 ETM+ TOA image, ground truth mask, and prediction made by a U-Net with a ResNet-18 backbone pre-trained using MoCo and SSL4EO-L and fine-tuned on L7 Irish.

Figure 5 shows an example prediction made by a U-Net pre-trained on SSL4EO-L and fine-tuned on L7 Irish. The model is able to correctly detect the majority of clouds in the image, but fails to detect cloud shadow due to its infrequent appearance in the training dataset. However, the model actually does a better job than the human annotator in the lower left corner, where the "ground truth" mask misses substantial cloud and thin cloud.

## 5 Limitations

There are a few limitations of the sampling method we chose to create our pre-training dataset. Due to low light levels near the poles, Landsat satellites do not capture images above 81.8° latitudes [77], and do not produce SR products above 76° latitudes.[4] The additional 23.5° tilt of the Earth's axis during the winter [78] means that it is not possible to collect imagery for all 4 seasons above 52.5° latitude. It may be possible to relax this constraint and allow for sampling from locations where 3 out of 4 seasons have imagery. Due to cloud cover and lower populations, there is very little imagery of tropical rainforests or polar regions, both of which are common applications of Landsat data.

The benchmark datasets we create are limited to the United States and may not adequately reflect performance in other regions where agricultural practices and crops differ greatly. Ideally, we would create additional global datasets. There exist large global Landsat-based datasets including the Global Forest Cover Change dataset [40]. However, these datasets do not exist during all times when these satellites are active. We would also like to have classification datasets in addition to semantic segmentation datasets. It may be possible to classify images by biome, although this task may be too easy. In future work, we would like to add pre-trained models for MSS data, although this will require a different sampling technique due to limited coverage over most of the world.

## 6 Conclusion

In this paper we introduce the SSL4EO-L pre-training dataset, the first ever SSL dataset for Landsat imagery and the largest Landsat dataset in history. We pre-train the first foundation models for the Landsat family of satellites, enabling progress in a multitude of scientific fields that can benefit from remote sensing and deep learning. Additionally, we revitalize the L7 Irish and L8 Biome datasets. We create the first benchmark datasets for the TM and ETM+ SR sensors, allowing direct comparison across all modern Landsat sensors and products. All datasets, model weights, training code, and scripts used to produce our results are distributed via the TorchGeo library, allowing for ease of experimentation and reproduction of our results.

---

[4]https://www.usgs.gov/landsat-missions/landsat-collection-2-surface-reflectance

## Acknowledgments and Disclosure of Funding

The authors gratefully acknowledge the computational and data resources provided through the joint high-performance data analytics (HPDA) project "terrabyte" of the German Aerospace Center (DLR) and the Leibniz Supercomputing Center (LRZ). This work was supported by the Helmholtz Association's Initiative and Networking Fund on the HAICORE@FZJ partition. This work made use of the Illinois Campus Cluster, a computing resource that is operated by the Illinois Campus Cluster Program (ICCP) in conjunction with the National Center for Supercomputing Applications (NCSA) and which is supported by funds from the University of Illinois at Urbana-Champaign. The work was supported in part by the National Science Foundation (NSF) through awards IIS 21-31335, OAC 21-30835, DBI 20-21898, as well as a C3.ai research award and the Taiwan-UIUC Fellowship.

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

# A    Appendix

## A.1    Ethics statement

Although satellite imagery in general can pose ethical concerns for surveillance and military applications, the imagery used to pre-train our models is low resolution (30 m/px) and cannot be used for such purposes. The primary applications Landsat imagery is useful for are Earth observation, including downstream tasks like climate change, agriculture, and ecology. While model training does contribute to greenhouse gas emissions, we believe that the benefits of such foundation models, especially their ability to reduce training demands for end users, outweigh these contributions.

## A.2    Licensing

All data used to create our datasets is released by the USGS under public domain, and may be used, shared, transferred, or redistributed without restriction. All datasets and models we create are released under a CC0 1.0 Universal license. All code, including training scripts and core Torch-Geo contributions, is released under an MIT license. The authors bear all responsibility in case of violation of rights.

## A.3    Downloading

All data, metadata, and pre-trained models used or created in this paper can be downloaded from https://huggingface.co/torchgeo, either manually or using TorchGeo (see Listing 1). Dataset images are stored in the widely used GeoTIFF format. These datasets and models will be maintained in perpetuity and may be improved over time. All datasets include dataset cards describing the dataset size, source, and license. All models include model cards describing the library used to load them, source, and license.

```python
from torchgeo.datasets import SSL4EOL

ds = SSL4EOL(root="data", split="oli_sr", download=True)
```

Listing 1: Example download script for the OLI SR split of the SSL4EO-L pre-training dataset.

## A.4    Reproducibility

Instructions to recreate the pre-training and benchmark datasets, results, or plots, can be found at https://github.com/microsoft/torchgeo/blob/releases/v0.5/experiments/ssl4eo/landsat/README.md. Listing 2 shows example code for pre-training on SSL4EO-L and fine-tuning/evaluating on our benchmark datasets, and can be modified to control other aspects of the training process or to train on a different sensor/product. The TorchGeo v0.5 release is the first release containing the datasets and models used and created in this paper. If you encounter any problems, please open an issue on GitHub and we will clarify the documentation.

```
from lightning.pytorch import Trainer
from torchgeo.datamodules import (
    SSL4EOLDataModule, SSL4EOLBenchmarkDataModule
)
from torchgeo.trainers import MoCoTask, SemanticSegmentationTask

# Pre-train on SSL4EO-L using MoCo
datamodule = SSL4EOLDataModule(split="oli_sr", seasons=2, download=True)
task = MoCoTask(model="resnet18", weights=True, in_channels=7)
trainer = Trainer(max_epochs=200)
trainer.fit(model=task, datamodule=datamodule)

# Fine-tune and evaluate performance
datamodule = SSL4EOLBenchmarkDataModule(sensor="oli_sr", product="cdl")
task = SemanticSegmentationTask(model="unet", backbone="resnet18")
trainer = Trainer(max_epochs=100)
trainer.fit(model=task, datamodule=datamodule)
trainer.test(model=task, datamodule=datamodule)
```

Listing 2: Example training script to pre-train and benchmark a model on SSL4EO-L.

## A.5 Class distribution

The benchmark datasets we use suffer from extreme class imbalance. Below are tables documenting the value, description, and percentage of each class in all datasets. Fill/background classes are ignored during training and are not considered when computing these statistics.

### A.5.1 Cloud detection datasets

Clear pixels cover more area than all other classes combined.

Table 3: Class distribution for cloud detection datasets.

| Value | Description | L7 Irish | L8 Biome |
|------:|-------------|---------:|---------:|
| 0 | Fill | - | - |
| 64 | Cloud Shadow | 0.7 | 1.5 |
| 128 | Clear | 66.1 | 50.5 |
| 192 | Thin Cloud | 10.2 | 14.7 |
| 255 | Cloud | 23.0 | 33.2 |

### A.5.2 SSL4EO-L benchmark datasets

The top 3 classes cover more area than all other classes combined. Only classes with > 1% area are considered during evaluation, the rest are mapped to the background class. TM data is downloaded from 2011, while ETM+ and OLI data is downloaded from 2019. The TOA and SR versions have the same geographic locations, and therefore the same class distribution.

Table 4: Class distribution for SSL4EO-L NLCD.

| Value | Description | TM | ETM+ | OLI |
|---|---|---|---|---|
| 0 | Background | - | - | - |
| 11 | Open Water | 2.4 | 2.2 | 2.3 |
| 21 | Developed, Open Space | 2.7 | 2.7 | 2.6 |
| 22 | Developed, Low Intensity | 1.7 | 1.7 | 1.7 |
| 31 | Barren Land (Rock/Sand/Clay) | 1.0 | 1.0 | 1.0 |
| 41 | Deciduous Forest | 9.2 | 9.2 | 8.8 |
| 42 | Evergreen Forest | 12.2 | 11.9 | 12.1 |
| 43 | Mixed Forest | 3.4 | 3.4 | 3.2 |
| 52 | Shrub/Scrub | 22.4 | 22.8 | 23.6 |
| 71 | Grassland/Herbaceous | 14.9 | 14.6 | 14.6 |
| 81 | Pasture/Hay | 6.2 | 5.9 | 5.8 |
| 82 | Cultivated Crops | 16.6 | 17.3 | 17.1 |
| 90 | Woody Wetlands | 4.5 | 4.4 | 4.3 |
| 95 | Emergent Herbaceous Wetlands | 1.6 | 1.5 | 1.6 |
| - | Other | 1.2 | 1.4 | 1.3 |

Table 5: Class distribution for SSL4EO-L CDL.

| Value | Description | TM | ETM+ | OLI |
|---|---|---|---|---|
| 0 | Background | - | - | - |
| 1 | Corn | 4.6 | 4.9 | 4.7 |
| 5 | Soybeans | 3.6 | 4.1 | 3.9 |
| 24 | Winter Wheat | 1.9 | 1.6 | 1.6 |
| 36 | Alfalfa | 0.9 | 1.1 | 1.2 |
| 37 | Other Hay/Non Alfalfa | 1.2 | 1.6 | 1.6 |
| 61 | Fallow/Idle Cropland | 1.4 | 1.9 | 1.8 |
| 111 | Open Water | 1.7 | 1.7 | 1.7 |
| 121 | Developed/Open Space | 3.3 | 2.9 | 2.8 |
| 122 | Developed/Low intensity | 1.4 | 1.5 | 1.5 |
| 131 | Barren | 1.1 | 1.1 | 1.1 |
| 141 | Deciduous Forest | 11.9 | 10.6 | 10.2 |
| 142 | Evergreen Forest | 13.3 | 12.7 | 12.9 |
| 143 | Mixed Forest | 1.5 | 3.2 | 2.9 |
| 152 | Shrubland | 22.4 | 24.2 | 25.0 |
| 176 | Grass/Pasture | 20.3 | 16.6 | 16.5 |
| 190 | Woody Wetlands | 3.9 | 4.2 | 4.1 |
| 195 | Herbaceous Wetlands | 1.3 | 1.4 | 1.5 |
| - | Other | 4.2 | 4.7 | 4.8 |

## A.6 Spectral bands

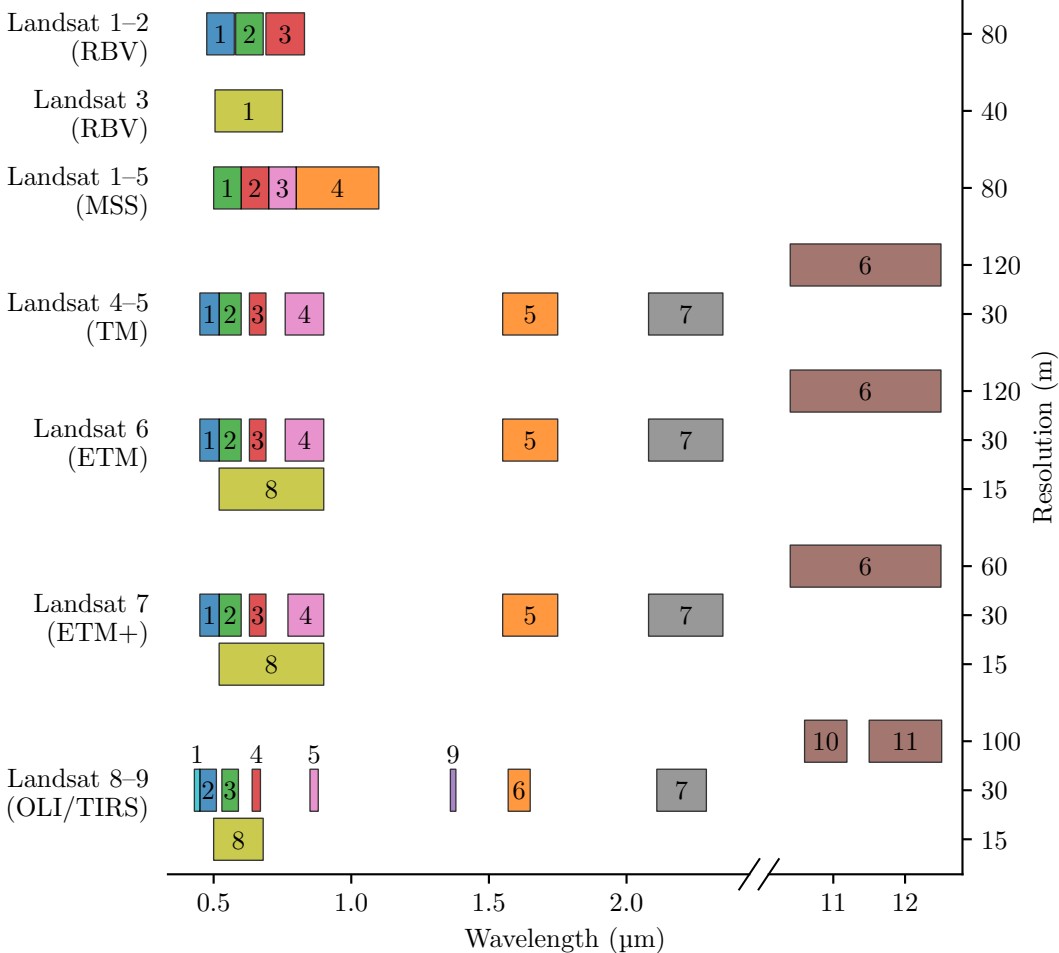

Figure 6: Spectral wavelengths and spatial resolutions of each band captured by all Landsat sensors. On Landsats 1–3, MSS bands were actually numbered 4–7. Landsat 9 introduced new and improved OLI-2/TIRS-2 sensors, but the bands are identical, so the sensors were combined in this figure.

## A.7 Data visualization

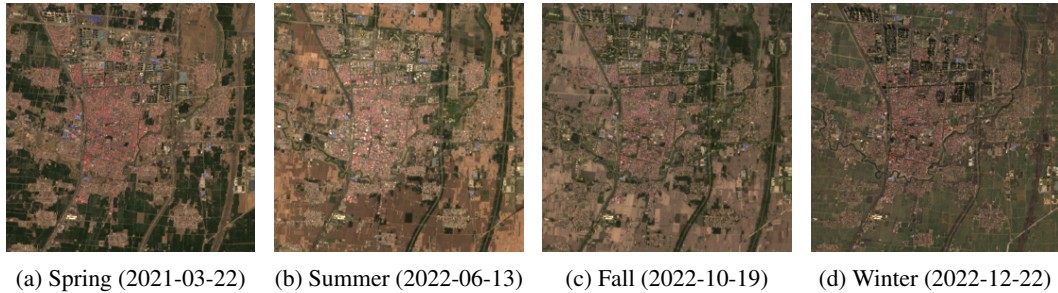

(a) Spring (2021-03-22)    (b) Summer (2022-06-13)    (c) Fall (2022-10-19)    (d) Winter (2022-12-22)

Figure 7: Example location showing the time-series nature of SSL4EO-L. Each location has imagery from 4 different seasons. Images are selected from a 60-day window centered about the vernal and autumnal equinoxes and the summer and winter solstices in order to maximize seasonal changes. Images are limited to a 2-year window to minimize man-made changes. Image location is Ci County, Handan, Heibei, China.

## A.8 Model complexity

Table 6: Complexity of backbone models used in this paper. Includes the number of parameters, memory requirements, floating point operations per second (FLOPS), and multiply-accumulate operations (MACs) of each model. All experiments were performed on an NVIDIA A100 GPU.

| Model | # Params (M) | Memory (MB) | FLOPS (G/s) | MACs (G) |
|-------|-------------|-------------|-------------|----------|
| ResNet-18 | 11.21 | 44.87 | 622.49 | 136.21 |
| ResNet-50 | 23.56 | 94.46 | 366.32 | 281.72 |
| ViT-S16 | 22.46 | 89.83 | 423.61 | 281.28 |

## A.9 Sampling algorithm

---

**Procedure** DownloadSSL4EO($N = 250,000, S = 4, \sigma = 50$ km)

---

**Data:** $M = \{\mu\}$ centroids of 10K most populous cities in the world
**Result:** Downloads non-overlapping, cloud-free, nodata-free images from $N$ locations during
$\quad\quad S$ seasons
$X \leftarrow \{\}$
**while** `len`$(X) < N$ **:**
$\quad \mu \sim \mathcal{U}(M)$
$\quad x \sim \mathcal{N}(\mu, \sigma)$

$\quad$`# Ensure x does not overlap with existing sampled patches`
$\quad$**if** `Overlaps`$(x, X)$**:** `# 264 px buffer`
$\quad\quad$**continue**

$\quad$`# Look for S cloud-free, nodata-free images at location x`
$\quad T \leftarrow \{\}$
$\quad t \leftarrow 0$
$\quad$**for** $t$ **in** `range`$(S)$**:** `# 60-day and 2-year window around equinoxes/solstices`
$\quad\quad$**if** `CloudCover`$(x, t)$**:** `# 20% threshold`
$\quad\quad\quad$**continue**

$\quad\quad$**if** `NoData`$(x, t)$**:**
$\quad\quad\quad$**continue**

$\quad\quad T \leftarrow T \cup \{t\}$
$\quad$**if** `len`$(T) < S$**:**
$\quad\quad$**continue**

$\quad$`# Download from Google Earth Engine`
$\quad$`Download`$(x, T)$
$\quad X \leftarrow X \cup \{x\}$

---

