# OpenReview forum: "SSL4EO-L: Datasets and Foundation Models for Landsat Imagery"
_NeurIPS.cc/2023/Track/Datasets_and_Benchmarks — NeurIPS 2023 Datasets and Benchmarks Poster_

### Official Review · Reviewer_JNXB · 2023-07-17
**Valuable dataset, interesting approach, flawed banchmarks**

**Rating:** 6
**Confidence:** 4

**Strengths:**

- the dataset is large (5M images) and cover the entire globe, which makes it a good resource to pre-train Earth Observation models.

- the authors propose 4 different benchmarks with distinct tasks

- the authors leverage existing data from previous EO missions and rejuvenated/fixed existing datasets. This is an interesting and frugal approach, which is commendable.

- the paper is generally well-written

**Additional Feedback:**

- In Tab1, it would be more interesting to report the results of a ResNet and Vit rather than 2 ResNets.

- The Irish and Biome results should be in the main paper, not just the appendix. Their results should also need to be analyzed.

**Clarity:**

- The writing is quite good, even if the authors overuse of terms that are not very academic, such as "myriad".

- More qualitative illustrations of the SL and benchmark data would clarify the tasks and nature of the acquisitions. While interesting and pleasant to read, a lot of technical data could be put in the appendix to make room.

**Correctness:**

>"an extensive literature review found almost no deep learning datasets for other sensors, products, or tasks"

What is a "deep learning dataset"?

**Documentation:**

Good

**Limitations:**

The authors addressed the limitations with interesting remarks

**Opportunities For Improvement:**

- I am dubious about the use of benchmarks whose ground truth labels are derived from automatic processing (using decision tree and random forest), and whose accuracy is very low. NLCD has an OA of 77.5%, close to the best algorithms' performance (~70%).
   - Wouldn't a random forest get close to 100%? In which case, what is the validity of training large deep learning models?
   - Such a low precision invalidates any macro-averaged metrics (mIoU), as some rare class could be completely mis-annotated

- Why is there 138 foundation models? I was not able to understand how this number came to be. Doesn't it completely defeat the purpose of *foundation* models?

> We replace this with an O (N log N ) R-tree [56], speeding up detection [sic] of patch

This is meaningless without an actual acceleration/time to gather the dataset.

- The authors should report the performance of models trained from scratch on the benchmark data to give perspective on the benefit of pre-training.

-The analysis of the results is lacking, as the authors merely report the results

**Relation To Prior Work:**

- the authors present an interesting retrospective on past Landsat missions and older benchmarks

**Summary And Contributions:**

The authors propose a unified dataset for LANDSAT, a series of multispectral Earth Observation satellites. They offer a curated, planet-spanning set of 5 million images across 5 generations of the satellite constellation. The authors also created 2 new dataset for evaluation purposes and fixed existing but forgotten/flawed datasets.

Models can use the dataset for self-supervision and be evaluated on the 4 proposed benchmarks.

---

> ### Author Response · Authors · 2023-08-18
>
> > I am dubious about the use of benchmarks whose ground truth labels are derived from automatic processing (using decision tree and random forest), and whose accuracy is very low. NLCD has an OA of 77.5%, close to the best algorithms' performance (~70%). Wouldn't a random forest get close to 100%? In which case, what is the validity of training large deep learning models? Such a low precision invalidates any macro-averaged metrics (mIoU), as some rare class could be completely mis-annotated
>
> We think there's a misunderstanding about how the ground truth labels (NLCD/CDL) are derived. You are correct that the ground truth labels are derived using decision trees and random forests. However, the input data is very different. In the benchmark datasets we create, the only input is a single snapshot image from Landsat taken during the summer months. However, NLCD and CDL products are generated using a time-series of imagery from several different satellites. For example, "The 2019 CDL product utilized satellite imagery from the Landsat 8 OLI/TIRS sensor, the Disaster Monitoring Constellation (DMC) DEIMOS-1 and UK2, the ISRO ResourceSat-2 LISS-3, and the ESA SENTINEL-2 sensors collected during the current growing season" [1]. Time-series information is known to provide significant boosts in performance for tasks like crop classification [2–6]. We also remove rare classes (classes with < 1% frequency) from our calculations to avoid the issues you describe. If all of our models were getting close to 77.5% OA, one could argue that the dataset is saturated and not useful for benchmarking. However, we don't see this, and the MoCo models consistently outperform ImageNet, allowing us to argue for their usefulness.
>
> [1] https://www.nass.usda.gov/Research_and_Science/Cropland/sarsfaqs2.php
> [2] https://doi.org/10.2134/agronj2007.0170
> [3] https://doi.org/10.1016/j.compag.2012.07.015
> [4] https://doi.org/10.1016/j.rse.2006.11.021
> [5] https://doi.org/10.1016/j.rse.2007.07.019
> [6] https://doi.org/10.1016/j.rse.2018.02.045
>
> > Why is there 138 foundation models? I was not able to understand how this number came to be. Doesn't it completely defeat the purpose of foundation models?
>
> Thanks for catching this, we think this is a case of poor terminology choice. Although there are 138 pre-trained models, we wouldn't describe all of them as "foundation models". The 138 number is calculated as follows.
>
> We have 5 sensors/products x 3 architectures x 2 SSL experiments for a total of 30 pre-trained backbones (excluding ImageNet because we didn't pre-train that ourselves). We also have 5 sensors/products x 3 architectures x 3 pre-training techniques x 2 tasks for a total of 90 pre-trained U-Nets. If you add in cloud detection that's another 3 architectures x 3 pre-training techniques x 2 datasets for another 18 pre-trained U-Nets. So in total, that makes 30 + 90 + 18 = 138 pre-trained weights.
>
> We ended up removing the 138 figure from the paper since the exact definition of "foundation model" is up for debate.
>
> > "We replace this with an O (N log N ) R-tree [56], speeding up detection [sic] of patch." This is meaningless without an actual acceleration/time to gather the dataset.
>
> Great point, and thank you for pointing this out. We tested this and it turns out there actually isn't a significant difference in speed between the R-tree and grid heuristic implementations. It's possible we're hitting the worst case time complexity instead of the average case. However, more importantly, the R-tree implementation is more technically correct and avoids the 1–3% overlap between patches reported by SSL4EO-S12. We’ve removed "speeding up detection of patch overlap" and instead focused on why using an R-tree instead of the heuristic prevents overlapping patches.
>
> > The authors should report the performance of models trained from scratch on the benchmark data to give perspective on the benefit of pre-training.
>
> Like most SSL papers, we chose to benchmark our pre-trained models by freezing the encoder and fine-tuning the decoder of a U-Net. This is similar to linear probing of classification models, and gives us a better idea of how the backbone will perform on datasets too small for full fine-tuning of models while requiring less computational resources for our 100+ fine-tuning experiments. Unfortunately we don't have sufficient computational resources to rerun all experiments with full fine-tuning, and it would not be a fair comparison to fully fine-tune a model with random weights and compare that to a frozen encoder that has never seen a supervised task before.

---

> > ### Author Response · Authors · 2023-08-18
> >
> > > The analysis of the results is lacking, as the authors merely report the results
> >
> > We completely agree with this comment. We've greatly expanded upon Section 4 to add more detailed analysis of our results.
> >
> > > "an extensive literature review found almost no deep learning datasets for other sensors, products, or tasks." What is a "deep learning dataset"?
> >
> > This was another unfortunate terminology choice. What we meant to say was benchmark datasets containing enough data points to train a deep learning model. We updated the paper to replace "deep learning dataset" with "benchmark dataset".
> >
> > > The writing is quite good, even if the authors overuse of terms that are not very academic, such as "myriad".
> >
> > We agree, and thank you for taking the time to point this out. The word "myriad" has been removed from the paper and replaced with quantifiable terms wherever possible. If you see any other ambiguous verbiage let us know.
> >
> > > More qualitative illustrations of the SL and benchmark data would clarify the tasks and nature of the acquisitions.
> >
> > Thanks for the suggestion. We added a figure in Appendix A.10 to provide additional visualizations, this time for one of our cloud detection datasets.
> >
> > > In Tab1, it would be more interesting to report the results of a ResNet and Vit rather than 2 ResNets.
> >
> > All ViT results have now been moved to Table 2 (formerly Table 1).
> >
> > > The Irish and Biome results should be in the main paper, not just the appendix. Their results should also need to be analyzed.
> >
> > All cloud detection benchmark results have been moved to Table 1 in the main paper. We've greatly expanded upon Section 4 to add more detailed analysis of the results of our cloud detection experiments.

---

> > ### Comment · Reviewer_JNXB · 2023-08-23
> > **Response to the rebuttal**
> >
> > I sincerely thank the authors for their thorough response and the improved manuscript.
> > Most of my concerns have been competently addressed, and I will upgrade my rating.
> >
> > Concerning the last point on fine-tuning, I understand that retraining from scratch 100+ model is not feasible. But it would be an interesting experiment for at least the best models. In many circumstances linear probing gives significantly worst results than fine-tuning.
> >
> > Alternatively, the authors can use LoRa [1] to speed up this fine-tuning.
> >
> > [1] LoRA: Low-Rank Adaptation of Large Language Models, Hu etal, ICLR22

---

### Official Review · Reviewer_awux · 2023-07-21
**The comments and suggestions on SSL4EO-L dataset paper.**

**Rating:** 6
**Confidence:** 4
**Clarity:** Yes.

**Strengths:**

SSL4EO-L developed the tile sampling strategy of satellite imagery compared the previous work, and first propose the SSL dataset and benchmark for the Landsat family of satellites. A large number of pre training models are open source for downstream tasks.

**Additional Feedback:**

No.

**Correctness:**

Yes, but some experimental results should to be reconfirmed, such as the result of SimCLR in Table 5 of supplementary material, where the range of standard deviation is large. Please reconfirm.

**Documentation:**

Yes.

**Ethics:**

No.

**Limitations:**

The related work in this paper is not detailed and the difference between SSL4EO-L and the previous work cannot be reflected in the Section 1.

**Opportunities For Improvement:**

1. line70-line71, it is suggested to highlight the differences between SSL4EO and previous self-supervised learning on the field of remote sensing. And the datasets and benchmark related to self-supervised learning of remote sensing should be summarized into a table in the Section 1.
2. line113-line116, please explain in detail the speeding up of patch samples, and suggest replacing steps 1)-6) with algorithm description or pseudocode to make it more understandable.
3. line124-line125, seasonal distribution is one of the characteristics of the dataset, but the authors do not emphasize it much, please add the description of the seasonal attributes in the dataset. For example, each tile of Landsat can be aligned the seasonal attributes for visualization in the Figure 3.
4. line251-line252, please describe the specific decoder architecture.
5. Table 1, It is recommended to add the results of ViT-Small and ViT-Base. In addition, the number of parameters and computational complexity of the model also need to be supplemented.
6. line264, the “Figure 1” should be modified into “Table 1”.
7. Figure 4, it is suggested to add images, visualization examples are too few in this paper and the supplements.


**Relation To Prior Work:**

Yes.

**Summary And Contributions:**

This paper proposed SSL4EO-L, the first ever dataset designed for self-supervised learning for Earth Observation for the Landsat family of satellites and the largest Landsat dataset in history, which performed excellently on downstream semantic segmentation tasks such as Landsat land cover classification, cloud detection, etc. This work has great potential for medium-resolution optical satellite sensors in global mapping applications.

---

> ### Author Response · Authors · 2023-08-18
>
> > line70-line71, it is suggested to highlight the differences between SSL4EO and previous self-supervised learning on the field of remote sensing. And the datasets and benchmark related to self-supervised learning of remote sensing should be summarized into a table in the Section 1.
>
> A detailed survey of prior work on SSL in RS is out of the scope of this paper, but we reference several review papers that readers can peruse to get up to speed on this fast moving field [1–3]. Differences between SSL4EO-L and prior iterations of SeCo and SSL4EO-S12 can be found in Section 2.
>
> [1] https://arxiv.org/abs/2206.13188
> [2] https://arxiv.org/abs/2204.02825
> [3] https://www.mdpi.com/2072-4292/14/16/3995
>
> > line113-line116, please explain in detail the speeding up of patch samples, suggest replacing steps 1)-6) with algorithm description or pseudocode to make it more understandable.
>
> We tested this and it turns out there actually isn't a significant difference in speed between the R-tree and grid heuristic implementations. It's possible we're hitting the worst case time complexity instead of the average case. However, more importantly, the R-tree implementation is more technically correct and avoids the 1–3% overlap between patches reported by SSL4EO-S12. We’ve removed "speeding up detection of patch overlap" and instead focused on why using an R-tree instead of the heuristic prevents overlapping patches. We added an algorithm version of the sampling technique to Appendix A.9.
>
> > line124-line125, seasonal distribution is one of the characteristics of the dataset, but the authors do not emphasize it much, please add the description of the seasonal attributes in the dataset. For example, each tile of Landsat can be aligned the seasonal attributes for visualization in the Figure 3.
>
> Seasonal Contrast as a data augmentation technique was introduced by SeCo. SSL4EO-S12 and our paper further refine the sampling technique used to create the pre-training dataset, but use seasonal attributes in the same way. The purpose of seasonal attributes is mentioned in Sections 2.1 and 3. We also added a figure in Appendix A.7 to visualize what these seasonal images look like.
>
> > line251-line252, please describe the specific decoder architecture.
>
> Great catch! We added the architecture (U-Net) to Section 3 and the captions of Tables 1–2.
>
> > Table 1, It is recommended to add the results of ViT-Small and ViT-Base. In addition, the number of parameters and computational complexity of the model also need to be supplemented.
>
> Results from ViT-Small have now been added to Tables 1–2. Unfortunately we don't have access to the computational resources required for ViT-Base. We also added a table in Appendix A.8 to compare the number of parameters, memory requirements, FLOPS, and MACs of each model.
>
> > line264, the “Figure 1” should be modified into “Table 1”.
>
> Good catch, fixed!
>
> > Figure 4, it is suggested to add images, visualization examples are too few in this paper and the supplements.
>
> Thanks for the suggestion. We added a figure in Appendix A.10 to provide additional visualizations, this time for one of our cloud detection datasets.
>
> > The related work in this paper is not detailed and the difference between SSL4EO-L and the previous work cannot be reflected in the Section 1.
>
> The difference between SSL4EO-L and SeCo/SSL4EO-S12 is discussed in Section 2.
>
> > some experimental results should to be reconfirmed, such as the result of SimCLR in Table 5 of supplementary material, where the range of standard deviation is large. Please reconfirm.
>
> We re-ran these experiments and triple checked the numbers and the standard deviations you see happen each time. Note that Table 5 in the earlier supplement is now Table 1 in the main paper.

---

> > ### Comment · Reviewer_awux · 2023-08-30
> > **Response to rebuttal**
> >
> > Thank you for your reply. I think the author has addressed most of my concerns.
> >
> > For self-supervised learning, there are many existing benchmark datasets in the field of remote sensing, please compare the size of the dataset, the size of the data volume, the data distribution, etc. and highlight the differences and advantages of your work in the introduction, otherwise it will be considered incomplete for comparison with previous work.

---

### Official Review · Reviewer_E2Pm · 2023-07-21
**SSL4EO-L:  Datasets and Foundation Models for Landsat Imagery**

**Rating:** 9
**Confidence:** 3
**Correctness:** Seems sound
**Clarity:** Yes

**Strengths:**

Great that the authors allowed for two sensors sampled from the same locations to allow for multi-senosr data fusion.

Great that the authors modernized and re-release the L7 Irish and L8 Biome cloud detection datasets as ML-reay data sets available for download from HuggingFace using the TorchGeo library. For example, they made sure that the issues with the existing versions are ironed out by replacing corrupted images and missing masks, resampling to 30m res, files were stacked and made into COGs, and made them available for automated download (rather than the manual download of the 206 and 96 files, respectively, availble previously).

**Additional Feedback:**

No additional feedback

**Documentation:**

An example of how to download and use the dataset/ getting started guide would be very useful.

**Limitations:**

Adding some of the points from above

**Opportunities For Improvement:**

The current city-centric sampling strategy ignores any place that has few cities or smaller ones biasing the dataset, which is only usable in the context it has been designed for i.e., large cities and their surroundings. Lines 100-107 uniform sampling is not the only other way that the authors can consider sampling but rather stratified random. Why not sample across LCLU classes and at the edges of those classes?

The authors should provide more details re the differences between thier data and SSL4EO-S12.

What do the authors mean by "TM SR and ETM+ SR are essentially identical, we did not create a separate dataset for TM SR." ? L5 and L7 are actually very different and there is substantial knowledge in the filed that mentions the issue w L5, such orbital drift, for ex: https://www.sciencedirect.com/science/article/pii/S0034425720300705

Line 209 - in RS you rarely or never use uniform sampling, you use stratified random sampling designs.

An example of how to download and use the dataset/ getting started guide would be very useful.

**Relation To Prior Work:**

Yes

**Summary And Contributions:**

The paper presents the Self-Supervised Learning for Earth Observation for the Landsat family of satellites (SSLEO-L), including 3 sensors and 2 product levels, and the largest Landsat dataset in history (5M image patches).

While other research is focused on more recent and higher resolution earth obervation (EO) data (e.g., Sentinel 2, Planet Scope), this paper focuses on the Landsat archive, which is very important in the context of historical research given that the data go back for 50y +. The dataset and model weights are distributed via the TorchGeo library.

---

> ### Author Response · Authors · 2023-08-18
>
> > The current city-centric sampling strategy ignores any place that has few cities or smaller ones biasing the dataset, which is only usable in the context it has been designed for i.e., large cities and their surroundings. Lines 100-107 uniform sampling is not the only other way that the authors can consider sampling but rather stratified random. Why not sample across LCLU classes and at the edges of those classes?
>
> This is a great suggestion and we completely agree. We based our methodology off of the proposed methods in SeCo [1] and SSL4EO-S12 [2]. The SeCo paper showed a consistent improvement in downstream task performance using a model that was pre-trained on a dataset of patches sampled around the most populated cities (i.e., a very similar method to this paper) vs. a dataset of patches collected by sampling uniformly at random (see Table 2). That said, the best sampling strategy is definitely an open question and one that deserves to be explored in greater detail in future work. For example, if we use stratified random sampling based on LCLU classes, which LCLU map works best? Some like CDL focus on agricultural classes and lump all non-agricultural classes into large buckets, while others differentiate between dozens of different forest biomes. Other heuristics we would be interested in include: stratified sampling over ecoregions, weighted sampling over gridded population data (e.g., Gridded Population of the World), diversity based sampling in some feature space, or simply continuously training on patches that are randomly sampled from entire archives (vs. pre-sampling locations).
>
> [1] https://arxiv.org/abs/2103.16607
> [2] https://arxiv.org/abs/2211.07044
>
> > The authors should provide more details re the differences between thier data and SSL4EO-S12.
>
> The main difference is that SSL4EO-S12 is for the Sentinel satellites and SSL4EO-L is for the Landsat satellites. Sensors onboard these satellites capture images with a different number of spectral bands, range of the electromagnetic spectrum, and spatial resolution, meaning datasets and pre-trained models are not compatible across sensors. We also improve on the sampling strategy of SeCo and SSL4EO-S12 by speeding up the patch sampling method and checking to avoid images containing nodata pixels. This is briefly mentioned in Section 2.1 but let us know if it should be made more explicit.
>
> > What do the authors mean by "TM SR and ETM+ SR are essentially identical, we did not create a separate dataset for TM SR." ? L5 and L7 are actually very different and there is substantial knowledge in the filed that mentions the issue w L5, such orbital drift, for ex: https://www.sciencedirect.com/science/article/pii/S0034425720300705
>
> You're absolutely correct, sensor issues including orbital drift can greatly affect the underlying data distributions in non-linear ways. This is part of the reason we chose to use L7 SR instead of L5 SR. What we meant to say is that the sensors themselves are identical (same number of spectral bands, wavelengths, and spatial resolutions) for the SR bands. This has been clarified in Section 2.1. In the case of orbital drift, the difference between new L5 and L7 is the same as the difference between old L5 and new L5, and the data distribution should actually look quite similar for old L5 and L7. In general, users should use caution when using imagery suffering from sensor issues including orbital drift.
>
> > Line 209 - in RS you rarely or never use uniform sampling, you use stratified random sampling designs.
>
> In the case of our benchmark datasets, due to the size of our patches and the number of locations that get skipped due to cloud cover or nodata pixels, 25K is already close to the limit of the number of patches that could be sampled from CONUS [1]. As we approach this limit, uniform random, stratified random, and city-centric Gaussian eventually select the same locations. We could sample fewer locations to achieve approximate class balance, but strong class imbalance is something that is observed in real-world land cover mapping, and the dataset can be used to evaluate performance on real-world data distributions.
>
> [1] Addendum: The area covered by our dataset is $25\textrm{K} \cdot (7.92\textrm{ km})^2 = 1.6\textrm{M km}^2$. The land area of CONUS is $8.1\textrm{M km}^2$. If we used a grid sampling technique, we could fill the entire area. However, the random sampling technique we use will only fill $\frac{1}{4}$ of the total area in the worst case. Cloud cover and nodata pixels explain the rest of the gap.
>
> > An example of how to download and use the dataset/ getting started guide would be very useful.
>
> Great idea, we added Listing 1 to Appendix A.3 to show how to download the dataset and Listing 2 to Appendix A.4 to show how to use it to reproduce the results in this paper.

---

### Official Review · Reviewer_XxEb · 2023-07-21
**Good paper, but some descriptions could be improved**

**Rating:** 8
**Confidence:** 4

**Strengths:**

The SSL4EO-L pre-training dataset is a large and carefully-prepared dataset for SSL. The dataset covers populated areas of the Earth. Creating the collection of patches with four different seasons with small cloud coverage is time-consuming, so this will help other researchers who are planning to apply the SSL technique for the Landsat data. The dataset archaeology for revitalizing L7 Irish and L8 Biome datasets is an additional contribution for evaluation, but its impact may be less than the SSL4EO-L benchmark dataset consisting of NLCD and CDL datasets. The strength of this paper is to create both pre-training and benchmark datasets for evaluating the effectiveness of SSL approaches to Landsat data.

**Additional Feedback:**

If the space allows, it is interesting to know a variety of tasks that the dataset may be helpful. Currently, the task is limited to semantic segmentation, but the dataset seems useful beyond those tasks.

**Clarity:**

The role of L7 Irish and L8 Biome datasets is not clear. Evaluation by NLCD and CDL datasets are discussed in the main part of the paper, and the L7 Irish and L8 Biome datasets are discussed in the appendix. Although the authors spent much effort revitalizing those datasets, it seems to be treated as datasets of second importance, and it is unclear why the authors decided to revitalize the L7 and L8 datasets. If those datasets serve different purposes from NLCD and CDL datasets, clarify those purposes to differentiate the purpose of the two types of datasets.

**Correctness:**

The reviewer could not find a description of how cloud coverage is computed. Is it obtained before downloading the data, or is it computed on the downloaded data for quality control? The 20% cloud coverage threshold is an essential parameter, and its accuracy impacts the dataset's quality.


**Documentation:**

The dataset is well organized, available from standard repositories. There is no description of the maintenance. Reproducibility is considered using a fixed random seed.

**Ethics:**

The reviewer does not observe any ethical concerns.

**Limitations:**

The authors admit the lack of global datasets that can be used as ground truth, but it is unclear why they avoid regional datasets for evaluation. If pre-training models are globally effective, they should work with regional datasets where the ground truth is created for the target region. Applying the model with adaptation to regional studies may significantly impact real-world applications.


**Opportunities For Improvement:**

The paper could be improved in terms of the benchmark results by clarifying the author's findings from the results. First, what is the implication of better performance of MoCo over ImageNet? Second, what is the implication of better performance of MoCo over SimCLR?

Moreover, the paper should emphasize the value of the pre-training dataset, which seems to be the most significant contribution of the paper.
For example, the results in Table 1 are a mixture of many effects, and it is not easy to extract the effect of using the pre-training dataset. Something like an ablation study is preferred to evaluate the value of the dataset properly.


**Relation To Prior Work:**

The dataset uses similar algorithms to previous work, but the dataset created in this paper is different from previous work.

**Summary And Contributions:**

The paper proposes the Landsat dataset for self-supervised learning (SSL). The central datasets are the SSL4EO-L pre-training dataset and the SSL4EO-L benchmark dataset, the largest dataset in this theme. The author also curated existing datasets, the L7 Irish and L8 Biome datasets, for machine learning purposes. The authors also created pre-trained models for SSL4EO-L datasets and showed that MoCo SSL outperforms other SSL methods in many settings.

---

> ### Author Response · Authors · 2023-08-18
>
> > The paper could be improved in terms of the benchmark results by clarifying the author's findings from the results. First, what is the implication of better performance of MoCo over ImageNet? Second, what is the implication of better performance of MoCo over SimCLR?
>
> Thanks for the comment, we've greatly expanded upon Section 4 to add more detailed analysis. The better performance of MoCo/SSL4EO-L over ImageNet implies that our pre-trained models will also provide better performance on other downstream tasks, and should be preferred over generic ImageNet weights. The implication of better performance of MoCo over SimCLR is harder to tease out, and may be due to not yet finding the right hyperparameters for SimCLR. Although our MoCo pre-trained weights are clearly superior to our SimCLR weights, our results should not be used to imply that MoCo is superior to SimCLR in terms of SSL methodology. It's somewhat expected that MoCo v2 would perform better than SimCLR v1 since it came out more recently, and SimCLR is known to perform best on TPUs supporting large batch sizes, which we did not have access to.
>
> > the paper should emphasize the value of the pre-training dataset, which seems to be the most significant contribution of the paper. For example, the results in Table 1 are a mixture of many effects, and it is not easy to extract the effect of using the pre-training dataset.
>
> We agree that the pre-training dataset, and the pre-trained models that it enables, are the most significant contributions of the paper. The goal of Tables 1–2 is to demonstrate that models pre-trained on ImageNet (RGB) are inferior to models trained on SSL4EO-L (MSI). Even though large supervised datasets do not exist for Landsat, unsupervised datasets like SSL4EO-L enable powerful SSL techniques like MoCo and SimCLR to pre-train models for downstream fine-tuning. As SSL4EO-L is the first ever SSL dataset for Landsat, it's difficult to compare to anything other than ImageNet. SeCo [1] and SSL4EO-S12 [2] provide detailed ablation studies of the sampling method we use if you're interested in that.
>
> [1] https://arxiv.org/abs/2103.16607
> [2] https://arxiv.org/abs/2211.07044
>
> > The authors admit the lack of global datasets that can be used as ground truth, but it is unclear why they avoid regional datasets for evaluation.
>
> Not only is there a lack of global datasets, there is also a lack of regional or even local datasets for anything other than semantic segmentation of OLI/TIRS TOA imagery. Of the 20+ Landsat benchmark datasets we found, only 3 were for ETM+ TOA and 1 was for MSS (but not large enough for deep learning). We found no benchmark datasets for ETM+ SR or TM. We were forced to create our own regional benchmark datasets based on NLCD/CDL to gauge the performance of our pre-trained backbones.
>
> > The reviewer could not find a description of how cloud coverage is computed.
>
> Cloud coverage is based on the pixel QA band (calculated by CFMask [1]) of the entire scene, not the small patch we actually download. This may result in patches with > 20% cloud cover if we happen to sample from a corner of a scene where clouds are concentrated. This is the same technique used by SeCo and SSL4EO-S12. Although patch-based cloud cover percentages would be ideal, the current algorithm already took weeks to download all 5M patches, and computing our own cloud cover mask or computing the percentage of each patch during the download process would be too computationally expensive. See this script [2] if you want to see exactly how the process works.
>
> [1] https://www.usgs.gov/landsat-missions/cfmask-algorithm
> [2] https://github.com/microsoft/torchgeo/blob/main/experiments/ssl4eo/download_ssl4eo.py
>
> > The role of L7 Irish and L8 Biome datasets is not clear.
>
> The cloud detection benchmark results table has now been moved to the main text and more detailed analysis of these results has been added to Section 4. We also added additional visualizations and analysis of L7 Irish in Appendix A.10.
>
> > There is no description of the maintenance.
>
> This can be found in Appendix A.3:
>
> "These datasets and models will be maintained in perpetuity and may be improved over time."
>
> > If the space allows, it is interesting to know a variety of tasks that the dataset may be helpful. Currently, the task is limited to semantic segmentation, but the dataset seems useful beyond those tasks.
>
> Yes, our benchmark dataset is derived from NLCD and CDL and primarily intended for semantic segmentation. Although it is possible to aggregate the most common land cover classes in each patch for multi-label classification, this task is much easier than semantic segmentation so we didn't evaluate it in our paper. It's also possible to use the parallel corpus for multimodal data fusion studies (e.g., what happens when your input is TOA & SR, or TM & ETM+ & OLI/TIRS?) or ablation studies (e.g., what happens when you train on SR and test on TOA?).

---

> > ### Comment · Reviewer_XxEb · 2023-08-31
> >
> > Thank you for your comments. It helped me understand the paper better.

---

### Official Review · Reviewer_hrbr · 2023-07-22
**Very good dataset on SSL for EO, with minor limitations**

**Rating:** 7
**Confidence:** 3
**Correctness:** yes
**Clarity:** Very well written.

**Strengths:**

The dataset is built in a very sound fashion. The efforts to collect old Landsat product are laudable

**Additional Feedback:**

Line 264: "Table 1" instead of Figure 1.

**Documentation:**

Very well presented.

**Limitations:**

One limitation that is not captured by the authors in the paper is the use of US-limited products for the evaluation of downstream tasks. It would be more informative to have a global product (https://land.copernicus.eu/global/products/lc) to assess how good is the generalization ability of the models.

**Opportunities For Improvement:**

Not clear what results Table 1 collects among the train-val-test results. Moreover, the benchmark without SSL pertaining is not available, so not clear how much SSL is actually useful. The pertaining seems to be performed on the whole dataset, and so it is not clear if the pertaining is done according to the same train-val-test split of the supervised task or not. Also, it would be nice to see if only pertaining in the US, where the products for supervised learning are available, would give all the benefits of the pertaining according to the whole dataset.
Lastly, a random train-val-test split should be used only in addition to proper spatial-informed splits, since the former can strongly overestimate the generalization ability of the models.

**Relation To Prior Work:**

Yes.

**Summary And Contributions:**

The paper develops a dataset designed for Self-Supervised Learning for the Landsat family of satellites. Additionally, the authors introduce ML benchmark datasets for Landsats 4–5 TM and Landsat 7 ETM+ SR. Finally, they develop foundation models for Landsat imagery using the dataset and SOTA SSL methods and evaluate their performance on two downstream semantic segmentation task.

---

> ### Author Response · Authors · 2023-08-18
>
> > Not clear what results Table 1 collects among the train-val-test results.
>
> We have significantly expanded the captions of Tables 1–2 to clarify that the results shown are from the test sets of the respective datasets.
>
> > the benchmark without SSL pertaining is not available, so not clear how much SSL is actually useful.
>
> Tables 1–2 present experimental results from both backbones pre-trained on ImageNet (without SSL) and backbones pre-trained on SSL4EO-L (with SSL). In this case, any gains over ImageNet can be attributed to in-domain pre-training with SSL.
>
> > The pertaining seems to be performed on the whole dataset, and so it is not clear if the pertaining is done according to the same train-val-test split of the supervised task or not.
>
> There seems to be some confusion regarding the difference between our pre-training datasets (1M images) and benchmarking datasets (25K images). The pre-training datasets contain only images, and the entire pre-training dataset is used to pre-train each backbone. The benchmarking datasets contain both images and masks, and are used to fine-tune and evaluate the performance of each U-Net using a 70-15-15 train-val-test split. The benchmarking datasets are not a subset of the pre-training datasets, and are collected from different years than our pre-training datasets, preventing spatiotemporal overlap. This is discussed in Section 2.3 of the paper.
>
> > it would be nice to see if only pertaining in the US, where the products for supervised learning are available, would give all the benefits of the pertaining according to the whole dataset.
>
> SSL4EO-S12 [1] actually performed this exact experiment, see Tables V, XVIII, and XIX. The authors found that pre-training on SSL4EO-S12 (global) resulted in better performance than pre-training on BigEarthNet (Europe) when evaluated on EuroSAT (Europe).
>
> [1] https://arxiv.org/abs/2211.07044
>
> > a random train-val-test split should be used only in addition to proper spatial-informed splits, since the former can strongly overestimate the generalization ability of the models.
>
> This is a great point. It's true that the test metrics reported in our paper may not accurately reflect how well our pre-trained models would perform on downstream tasks in new geographic regions. However, the same random train-val-test split was used to compare ImageNet, MoCo, and SimCLR. Therefore, the fact that MoCo consistently outperforms ImageNet still suggests that SSL4EO-L foundation models are superior to ImageNet foundation models for Landsat imagery. The fact that our models work so well on cloud detection despite a sampling method and pretext task explicitly designed to ignore clouds also demonstrates the generalization ability of the models to new tasks. Discussion on this latter style of generalizability was added to Section 4. A more detailed transfer learning study, including not only geographic transfer but also transfer between different application domains, is left for future work.
>
> > One limitation that is not captured by the authors in the paper is the use of US-limited products for the evaluation of downstream tasks. It would be more informative to have a global product (https://land.copernicus.eu/global/products/lc) to assess how good is the generalization ability of the models.
>
> This is already mentioned in our Limitations section:
>
> "The benchmark datasets we create are limited to the United States and may not adequately reflect performance in other regions where agricultural practices and crops differ greatly. Ideally, we would create additional global datasets. There exist large global Landsat-based datasets including the Global Forest Cover Change dataset. However, these datasets do not exist during all time periods when these satellites are active."
>
> The Copernicus Land Cover dataset you reference suffers from the same issue, it didn't exist before 2015. It also has a spatial resolution of 100 m instead of 30 m.
>
> > Line 264: "Table 1" instead of Figure 1.
>
> Good catch, fixed!

---

### Decision · Program_Chairs · 2023-09-22

**Decision:**

Accept (Poster)

**Comment:**

* Summary. The manuscript presents the Landsat program's extensive history and introduces SSL4EO-L, a groundbreaking dataset for self-supervised learning in Earth observation, featuring 5 million image patches. It also modernizes existing datasets and presents benchmark datasets for Landsat 4-8. The paper pioneers the pre-training of foundation models for Landsat imagery and offers open access to datasets and model weights via the TorchGeo library, facilitating reproducibility and scientific advancements in remote sensing.
* On reviewer’s comments. Reviewers generally found the manuscript promising, but they also identified specific areas for improvement. Reviewers 1 and 2 both highlighted the strengths of the SSL4EO-L dataset, emphasizing its size and quality. Reviewer 2 particularly praised the effort to modernize and release datasets like L7 Irish and L8 Biome. However, both reviewers pointed out opportunities for clarification and deeper analysis. Reviewer 1 questioned the clarity of Table 1's results and the necessity of a non-SSL benchmark. They also recommended addressing spatial-informed splits and exploring the benefits of SSL only in the US. Reviewer 2, on the other hand, suggested clarifying the implications of MoCo's performance and conducting an ablation study to evaluate the dataset's impact properly. Reviewer 3 appreciated the manuscript's historical context, focusing on the Landsat archive's significance and the distribution of the dataset via the TorchGeo library. They also acknowledged the inclusion of two sensors sampled from the same locations for multi-sensor data fusion. However, this reviewer recommended exploring alternative sampling strategies and providing more details about the differences between SSL4EO-L and previous work in remote sensing. Finally, Reviewer 5 noted the strengths of the large SSL4EO-L dataset and the efforts to rejuvenate existing datasets. They also raised concerns about the benchmarks' ground truth labels and questioned the necessity of numerous foundation models. Additionally, they sought clarification on specific technical details, such as cloud coverage computation, and recommended reporting the performance of models trained from scratch. Overall, reviewers appreciated the manuscript's contributions and provided constructive feedback to enhance clarity, analysis, and the dataset's utility.
* Author’s responses. The authors engaged in a fruitful and constructive discussion and provided convincing explanations to the reviewer’s comments, addressing most of their concerns. After the multiple review comments, the general feeling is that most problems have been discussed competently, and some reviewers even upgraded the initial rating.
* On this work's quality, clarity, originality, and significance. This work demonstrates high quality in its meticulous construction of the SSL4EO-L dataset, the modernization of existing Landsat datasets, and the development of foundation models and benchmarks for Landsat imagery. The manuscript is clearly written, although further opportunities exist to clarify specific implications and methodologies. The originality of this research lies in its focus on the historical Landsat archive, providing a valuable resource for the remote sensing community and addressing a gap in self-supervised learning for Earth observation. Furthermore, the significance of this work is notable as it not only contributes to the largest Landsat dataset in history but also offers a foundation for advancing remote sensing applications and scientific research, particularly for historical Earth observation data, through its open-access distribution and benchmark evaluations.
Suggestion. Clear accept.